# An unbiased approach to measure aberrant DNA methylation alterations

Bradley M. Downs [1] ✉, Jiumei Hu[2], Joon Soo Park [3], Hanran Lei[4], Tza-Huei Wang [1,2,4], Thomas R. Pisanic II [1], Kuangwen Hsieh [2] & Tra My Hoang[5]

The ability to accurately measure aberrant DNA methylation levels is integral to the understanding of DNA methylation biology. It is well-established that in cancer, the largest, and thus, most biologically important absolute gains of DNA methylation levels occur at CpG sites with low native levels while the largest losses occur at CpG sites with high native levels. Conventional wisdom assumes that the observed association between the degree of the alterations and the native levels are largely due to the limitations of change within the DNA methylation scale. Here, we present evidence that this association is largely caused by alterations occurring as a global rate of change relative to the native level. We show that DNA methylation alterations can be accurately compared by calculating the rate of change relative to the native level. Most importantly, this approach enables the identification of more biologically significant DNA methylation alterations.

Methylated DNA can be found in plants, animals, fungi, and bacteria, and is a key epigenetic regulator of gene expression, and thus, is essential for normal cell function[1]. In general, DNA methylation studies are conducted by comparing the absolute difference in DNA methylation levels of CpG sites, which are commonly measured as the percentage or proportion (Beta value) of cytosine residues that are methylated. Studies that identify DNA methylation alterations of CpG sites often do so by using two characteristics: the probability that the DNA methylation levels are statistically different ($P$ value) and the absolute degree of difference between DNA methylation levels (ΔMeth), which can be used as an indicator of biological significance[2–4]. The most basic method to calculate ΔMeth is to subtract the DNA methylation levels of CpG sites between samples, where a larger ΔMeth value assumes a more biologically significant change. However, due to multiple factors, including that DNA methylation levels can only range between 0.0–1.0 and CpG sites are naturally methylated at specific levels (which are required for cellular differentiation and proper cellular function)[5–7], the assumption underlying the usage of ΔMeth may not always hold true.

Modifications of the epigenome, including DNA methylation alterations at CpG dinucleotides, have been found to occur in many diseases and are one of the earliest and most frequent aberrant changes in cancer[8,9]. It is well-established that many of the largest gains of DNA methylation (hypermethylation) in cancer are in CpG Islands, which are largely unmethylated in normal cells and have been shown to lead to gene silencing of tumor suppressors, and thus, contributes to the basic biological understanding of the disease[10–12]. Early studies were perplexed about the biological importance of the loss of DNA methylation (hypomethylation), which has been found to occur in genes in confounding biological pathways for cancer progression[13]. However, more recent studies have shown that large global hypomethylation events at CpG sites that are normally heavily methylated induce chromosomal instability through an increased rate of DNA double-strand breaks and the reactivation of transposable elements[14–16].

Conventional wisdom assumes that the observed association between the degree of ΔMeth and the native level of the CpG sites are largely due to limitations of change within the DNA methylation scale

[1]Institute for NanoBioTechnology, Johns Hopkins University, Baltimore, MD, USA. [2]Department of Mechanical Engineering, Johns Hopkins University, Baltimore, MD, USA. [3]Department of Chemistry, Konkuk University, Seoul, Republic of Korea. [4]Department of Biomedical Engineering, Johns Hopkins University, Baltimore, MD, USA. [5]MilliporeSigma, Rockville, MD, USA. ✉e-mail: bradley.downs1@gmail.com

(0.0–1.0). For example, the maximum possible loss of DNA methylation of a CpG site with a native level of 0.80 is larger than that of a CpG site with a native level of 0.10. However, since DNA methylation alterations can be caused by factors that affect the whole methylome, such as age, environmental exposures and genetic alterations[17–19], it is conceivable that the CpG sites of the methylome undergo a similar global rate of change relative to their native level (relative ΔMeth). For example, a 50% global loss of DNA methylation relative to the native level would change a CpG site with a native level of 0.80 to 0.40 and a CpG site with a native level of 0.10 to 0.05. Although both scenarios would cause the degree of ΔMeth to be associated with the native level, whether ΔMeth is an inherently biased approach for measuring the level of the DNA methylation alteration depends on the nature at which alterations occur: as an absolute level change or a relative level change.

If the DNA methylation alterations occur as a global rate of change relative to the native level, then it reasons that the ΔMeth calculation would be inherently biased based on the native level. Since the majority of current approaches, including the ΔMeth calculation, assume that alterations occur as an absolute level change, the current understanding of DNA methylation would also be biased. Therefore, determining the nature at which DNA methylation alterations occur, and using an appropriate approach to measure the level of the alteration, is fundamental for the understanding of DNA methylation biology.

Here, we propose an alternative hypothesis, that the association between the degree of ΔMeth and the native levels are due to DNA methylation alterations occurring at similar global rates of change relative to the native level. Thus, we propose that the selection for large relative ΔMeth would be an unbiased approach to enrich for more biologically important alterations in comparison to those identified using ΔMeth. To test this hypothesis, we calculated and compared the features of ΔMeth and the relative ΔMeth using a comprehensive dataset of cancer tissues from The Cancer Genome Atlas (TCGA) and a large diverse dataset of normal tissues. We found multiple lines of evidence suggesting that DNA methylation alterations occur at similar global rates of change relative to the native level and that the selection for large ΔMeth skews for CpG sites based on their native level. We also found that the relative ΔMeth can more accurately identify DNA methylation signatures and can uniquely detect DNA methylation signatures caused by multiple factors including genetic

alterations, smoking and age, in comparison to ΔMeth. More significantly, we found that large relative ΔMeth identified genes in pathways that are more biologically important to cancer development and progression in comparison to ΔMeth. This result was most striking in the hypomethylation direction. While we found that the CpG sites with large ΔMeth in the hypomethylation direction were enriched in pathways important for brain function, a commonly identified and confounding pathway for cancer development, we found that the CpG sites that have large relative ΔMeth were enriched in multiple pathways important for tumor progression, including pathways for cell adhesion, increased metabolism, cell signaling and immune activation. Thus, we conclude DNA methylation alterations generally occur at a global rate of change relative to the native level and that the relative ΔMeth calculation is an unbiased approach that can identify more biologically important alterations, compared to ΔMeth.

## Results

Although it is commonly observed that there is an association between the degree of ΔMeth and the native level, why this association occurs remains unknown. To determine whether this association is caused by alterations occurring as a global rate of change relative to the native level, we calculated the ΔMeth and relative ΔMeth at CpG sites with varying native levels in a large and comprehensive dataset of cancers. To highlight the difference between ΔMeth and relative ΔMeth, an illustration showing examples of calculating, comparing, and the analysis of the two approaches at CpG sites of varying native DNA methylation levels is shown in Fig. 1.

### Global methylation changes in cell lines

To demonstrate how the association between the degree of ΔMeth and the native DNA methylation level can be caused by alterations occurring at a similar global rate of change relative to the native levels, we investigated the 5-azacytidine induced global loss of DNA methylation in multiple cell lines (3BKO and 3ABDKO) that have deficiencies in regulating DNA methylation because of a deletion of one or multiple DNA methyltransferase (DNMT) genes[17]. Studies have shown that 5-azacytidine causes global DNA methylation loss by incorporating into the DNA, replacing the cytosine nucleotides and inhibiting DNMTs[20]. To investigate the association between the degree of ΔMeth and the native levels, we subset the CpG sites of the Illumina Human

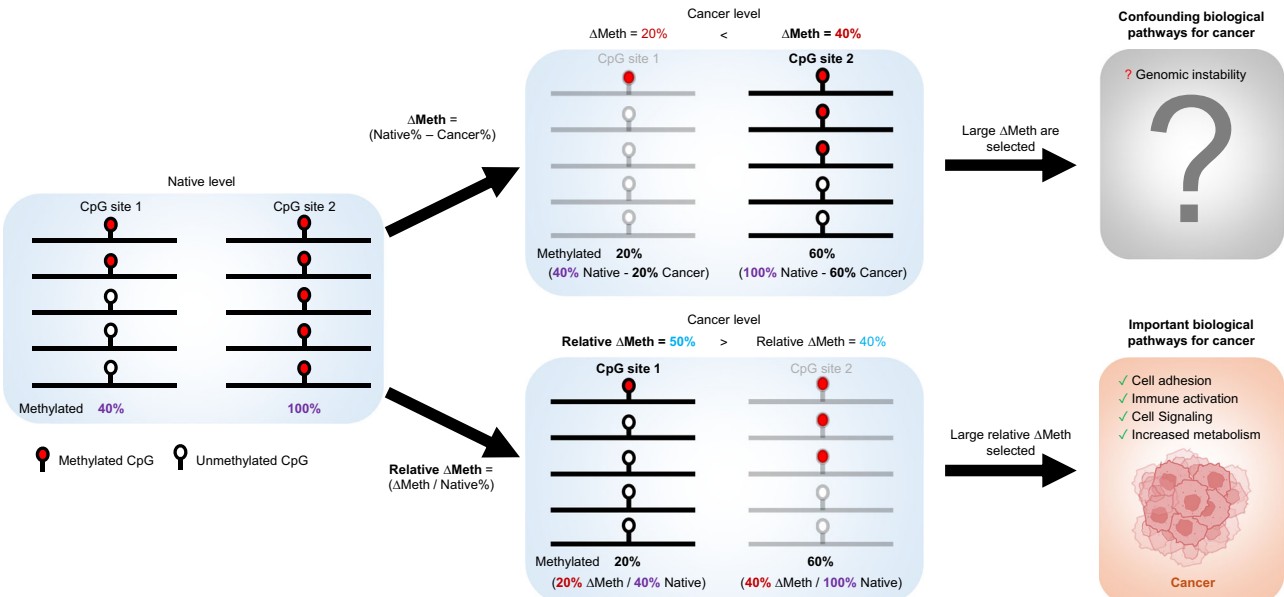

**Fig. 1 | Illustration of the ΔMeth and relative ΔMeth approaches.** Examples of calculating, comparing and analysis of hypomethylation alterations at CpG sites with varying levels of native DNA methylation using ΔMeth and relative ΔMeth.

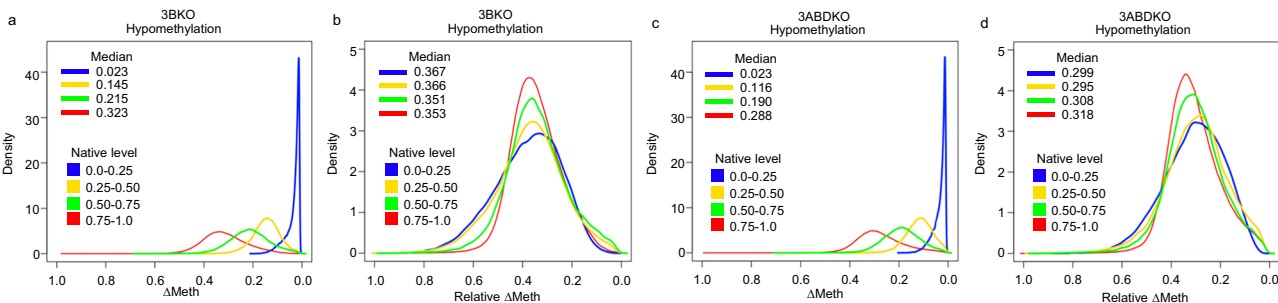

**Fig. 2 | Distribution profiles of ΔMeth and relative ΔMeth in 5-azacytidine treated cell lines.** Density plots showing the distribution of hypomethylation alterations in 3BKO cells using (**a**) ΔMeth and (**b**) relative ΔMeth and in 3ABDKO cells using (**c**) ΔMeth and (**d**) relative ΔMeth.

Methylation 450 K array datasets (3BKO; GSE51815 and 3ABDKO; GSE68344) into four even quantile intervals based on the untreated cells' DNA methylation level (native level). We considered the DNA methylation level of the treated cells to be changed if the difference between the untreated and treated cells (ΔMeth) was ≥0.01. We found in both cell lines after 5 days of 5-azacytidine treatment, in the hypomethylation direction, that the degree of ΔMeth was positively associated with the native level, and the largest absolute degree of ΔMeth occurred at CpG sites with the highest native levels (Fig. 2a, c). Conversely, we found that the relative ΔMeth (ΔMeth/native DNA methylated level) was very similar across CpG sites with varying native levels (Fig. 2b, d). The median global loss of DNA methylation was 35–37% relative to the native level in the treated 3BKO cells and 30-32% in the treated 3ABDKO cells. These results illustrate how a similar global rate of DNA methylation change relative to the native level can lead to an association between the degree of ΔMeth and the native levels.

## Global DNA methylation changes in tumors

Next, we investigated whether CpG sites with varying native levels in tumor tissues also undergo similar global rates of DNA methylation change relative to the native level. While it is ideal to calculate the native level from matched tumor adjacent normal tissues, matched tissues are seldomly available. Thus, we used the OutlierMeth package, which was designed to overcome the limited availability of normal tissues[21], to calculate the native DNA methylation levels at the upper and lower 99th quantiles from Illumina Human Methylation 450 K array data derived from 25 normal tissue types consisting of 1991 samples (Supplementary Data Table 1). We used functions in the OutlierMeth package to calculate ΔMeth and relative ΔMeth. We considered the cancer DNA methylation level to be aberrant if it was greater than the upper native level or less than the lower native level. The equation (|aberrant cancer DNA methylation level – native DNA methylation level|) was used to calculate the ΔMeth in the hypomethylation and hypermethylation directions. To calculate the relative ΔMeth in the hypomethylation and hypermethylation directions, we used the equations (ΔMeth/native DNA methylated level) and (ΔMeth/(native DNA unmethylated level)), respectively. To investigate whether the native levels associate with the degree of the alteration, we subset the CpG sites into four even quantile intervals based on their native level (Supplemental Fig. 1a–c). Because of their known significance, we also subset the CpG sites based on their genomic region (Supplemental Fig. 1d–h).

We then calculated the hypomethylation and hypermethylation ΔMeth and relative ΔMeth from TCGA Illumina Human Methylation 450 K array datasets, which included 31 tumor types and 8,551 samples (Supplementary Data Table 2). To ensure all tumor types carried equal weight, each tumor type was analyzed separately prior to the results being combined. Density plots were then made to assess the levels and distribution profiles of the median ΔMeth at CpG sites with varying native levels of DNA methylation. To reduce the effects caused by

differences in the frequency of the aberrant events, we grouped the CpG sites based on the frequency of the alteration (number of samples with an aberrant DNA methylation level/total number of samples) before the median level was calculated (aggregated median). As expected, with the full set of CpG sites, in both the hypomethylation and hypermethylation directions, we found that the degree of ΔMeth was associated with the native DNA methylation level (Fig. 3a, b). Specifically, we found that, in the hypomethylation direction, the median ΔMeth increased as the native levels increased and, in the hypermethylation direction, the median ΔMeth decreased as the native levels increased. This trend was also observed in the individual tumor types and genomic regions, with the largest hypomethylation alterations occurring in OpenSeas and the largest hypermethylation alterations occurring in Islands (Supplemental Fig. 2 and Supplemental Fig. 3a, b).

Next, we calculated the relative ΔMeth in the hypomethylation and hypermethylation direction. We found that the only obvious difference was at CpG sites with the highest native DNA methylation levels in the hypomethylation direction, which had much smaller relative ΔMeth (Fig. 3c). Later, we will show that this result is largely due to differences in the frequency of the aberrant events. All other relative ΔMeth distribution profiles in the hypomethylation and hypermethylation direction were remarkably similar, with the median global gain and loss of DNA methylation being 22-25% relative to the native DNA methylation level. Similar relative ΔMeth distribution profiles were also found when investigating the individual tumor types and the different genomic regions (Supplemental Fig. 3c, d and Supplemental Fig. 4). Again, we found that the largest relative ΔMeth hypomethylation alterations occurred in OpenSeas and the largest hypermethylation alterations occurred in Islands.

We then investigated whether the overlapping profiles of the relative ΔMeth were the result of random level changes that are simply limited by the DNA methylation scale (0.0–1.0), which is a common assumption used to explain the association between the degree of ΔMeth and the native levels. After we made a dataset of randomly generated values equivalent to the size of the compiled TCGA cancer dataset (N = 8,551), we found that the median relative ΔMeth was exactly 0.50 at CpG sites with varying levels of native DNA methylation (Supplemental Fig. 5a, b). We found that the TCGA relative ΔMeth distribution profiles were statistically smaller (Wilcox rank sum test P < 0.001) than the random dataset, providing evidence that the overlapping distribution profiles of the TCGA data are not the results of random DNA methylation level changes.

## Effects of events on methylation levels

Since the relative frequency of the alteration in tumors can affect the degree of ΔMeth and relative ΔMeth, which would affect the distribution profiles (Fig. 3a–d), we controlled for this variable by calculating the rate between the degree of the alteration and the relative frequency of the alteration. To ensure that all tumor types carried

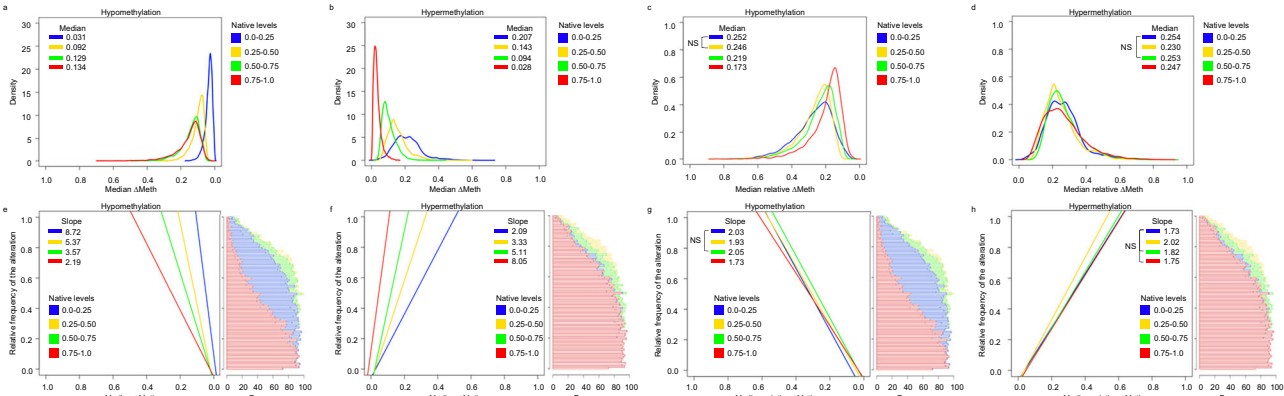

**Fig. 3 | Profiles of ΔMeth and relative ΔMeth in tumors at CpG sites with varying native levels.** Density plots showing the distribution of median ΔMeth in the (**a**) hypomethylation and (**b**) hypermethylation directions and the median relative ΔMeth in the (**c**) hypomethylation and (**d**) hypermethylation directions. Line plots showing the rate of change between the relative frequency of the DNA methylation alteration and the level of median ΔMeth in the (**e**) hypomethylation and (**f**) hypermethylation directions and with median relative ΔMeth in the (**g**)

hypomethylation and (**h**) hypermethylation direction. Histograms showing the frequency of the data based on relative frequency of the alteration. NS not statistically different ($P > 0.001$). All other distributions in each plot are statistically different (Wilcoxon rank sum test, two-sided $P < 0.001$) and all other slopes in each plot are statistically different (z-statistic, two-sided $P < 0.001$). All slopes are statistically significant (coefficient, two-sided $P < 0.001$).

equal weight, each tumor type was analyzed separately before the results were combined. Histograms were made to visualize the frequency of data with respect to the relative frequency of the alteration (Fig. 3).

From the TCGA dataset, as expected, we found that the aggregated median ΔMeth levels increased as the relative frequency of the alteration increased (coefficient $P < 0.001$), confirming the positive association between the frequency of the alteration and degree of ΔMeth (Fig. 3e, f). We also found that the rate of change was associated with the native level. In the hypomethylation direction, we found that the rate of change was largest at the highest native levels. And in the hypermethylation direction, the rate of change was largest at the lowest native levels. Thus, we conclude that there is an association between the degree of ΔMeth and both the relative frequency of the alteration and the native level. These two associations were also found when each tumor type was analyzed independently (Supplemental Fig. 6a, b).

Next, we calculated the rate of change between the relative ΔMeth and the relative frequency of the alteration. Similar to the ΔMeth, we found that the relative ΔMeth increased as the relative frequency of the alteration increased (coefficient $P < 0.001$) in both the hypomethylation and hypermethylation directions (Fig. 3g, h). However, unlike ΔMeth, we found that the relative ΔMeth rate of change was similar, ranging between 1.7–2.1, and was not associated with the native levels. Interestingly, the rate of change of the CpG sites with the highest native levels in the hypomethylation direction is lower than other rates of change, suggesting that the small distribution profile of these CpG sites (Fig. 3c) are largely due to differences in the relative frequency of the alterations (histogram of Fig. 3g). The variability of the rate of change in the different tumor types also seems to be quite similar throughout the varying native DNA methylation levels (Supplemental Fig. 6c, d).

Once again, we investigated whether the relative ΔMeth similarities are the result of random level changes being confined by the DNA methylation scale. Using the previously described dataset of randomly generated values, we calculated the rate of change between the relative ΔMeth and the relative frequency of the alteration. We found that the randomly generated dataset produces non-significant coefficients ($P > 0.05$) at CpG sites with varying levels of native DNA methylation (Supplemental Fig. 5c, d), again confirming that the similar rates of change between the relative ΔMeth and the relative frequency of the alteration are not due to random DNA methylation level changes.

## Detection of DNA methylation signatures

If DNA methylation alterations occur as a global rate of change relative to the native level, we would expect that the relative ΔMeth calculation would be more accurate than ΔMeth in detecting DNA methylation signatures at CpG sites with native levels close to the end of the DNA methylation scale (where the ΔMeth would be the smallest). To verify this prediction, we generated DNA methylation signatures in a dataset of 1000 randomly selected cancer samples from the compiled TCGA dataset. The DNA methylation signatures were created by altering a percentage (10, 20, 30, or 40%) of the CpG sites by a 10% gain or loss of DNA methylation relative to the cancer samples' native level. We also altered 10% of the CpG sites by a percentage (10, 20, 30, or 40%) of gain or loss of DNA methylation relative to the cancer samples' native level. As predicted, using the OutlierMeth package, we found that the aggregated median relative ΔMeth had higher accuracy in detecting hypomethylation signatures at CpG sites with low native levels and hypermethylation signatures at CpG sites with high native levels in comparison to ΔMeth (Supplemental Figs. 7, 8).

To determine if this phenomenon can also be found in TCGA cancer datasets, we calculated the aggregated median ΔMeth and relative ΔMeth to investigate multiple factors including genetic alterations, smoking, and age. Specifically, we investigated the well-characterized CpG Island methylator phenotype (CIMP) signature in colorectal cancers, smoking signature in lung cancers and the signature associated with age in breast cancers. In brief, colorectal CIMP tumors commonly carry a BRAF mutation and are characterized by having larger levels of hypermethylation in genomic CpG Islands at CpG sites with low native levels in comparison to non-CIMP tumors[22–24]. As expected, we found that high CIMP (H-CIMP) tumors had larger degrees of ΔMeth and larger relative ΔMeth levels of hypermethylation at CpG sites with low native levels in comparison to the non-H-CIMP tumors (Fig. 4a). However, at CpG sites with the highest native levels, only the relative ΔMeth detected a statistical difference between H-CIMP and non-H-CIMP tumors (Fig. 4b). Similarly, only the relative ΔMeth could detect hypomethylation signatures at CpG sites with the lowest native levels and hypermethylation signatures at CpG sites with the highest native levels between smokers and non-smokers in lung cancer (Fig. 4c, d). This was also observed in the signatures associated with age in breast cancer (Fig. 4e, f).

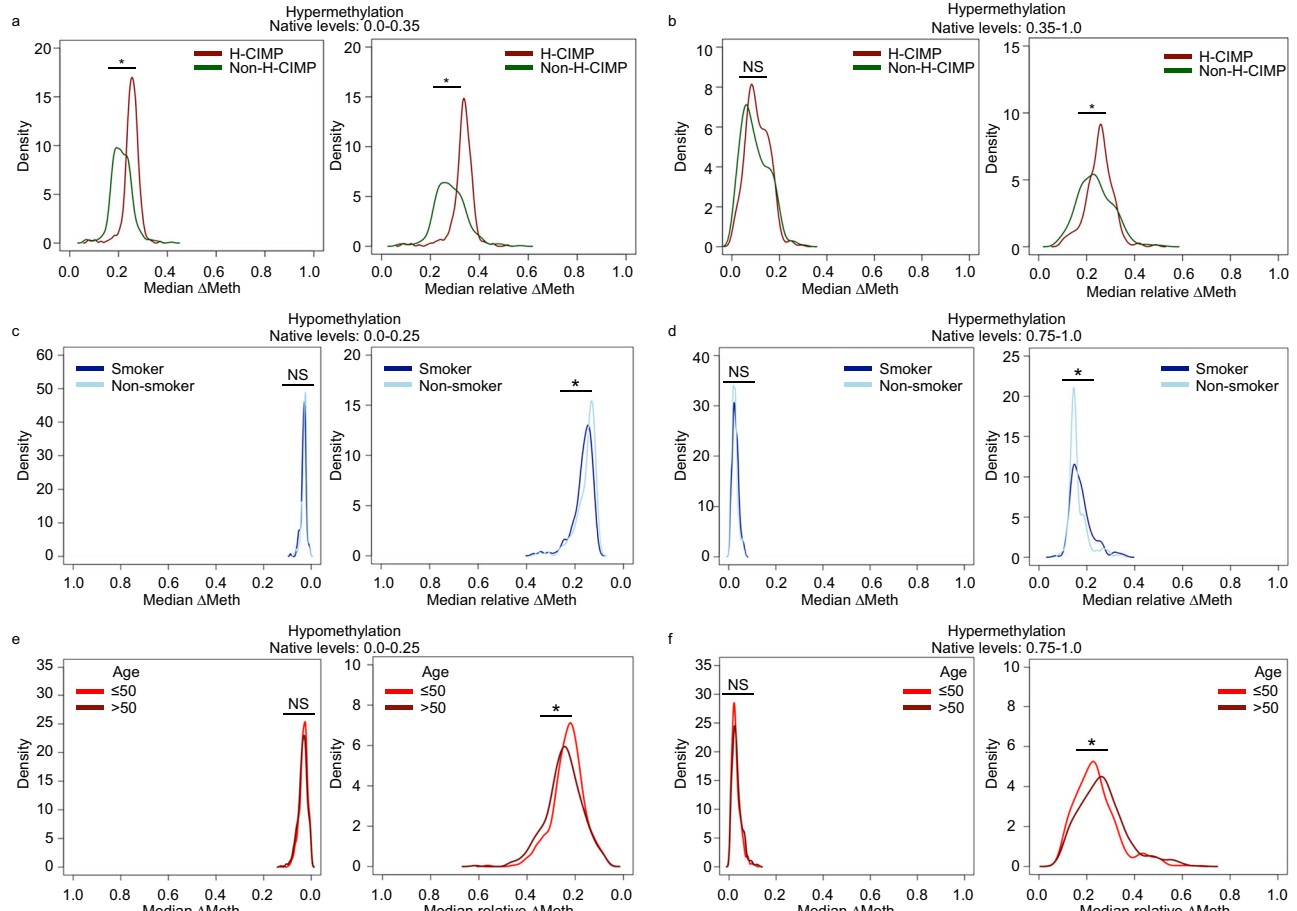

**Fig. 4 | Detection of DNA methylation signatures using ΔMeth and relative ΔMeth.** Density plots showing the distribution profiles of the median ΔMeth and relative ΔMeth in H-CIMP and non-H-CIMP colorectal cancer in the hypermethylation direction at CpG sites with (**a**) low native levels and (**b**). high native levels. Plots showing the distribution of the median ΔMeth and relative ΔMeth in smokers and non-smokers with lung cancer in the (**c**) hypomethylation direction in CpG sites with low native levels (ΔMeth $P = 0.002$, relative ΔMeth $P = 9.0E-05$) and in the (**d**) hypermethylation direction in CpG sites with high native

levels (ΔMeth $P = 0.005$, relative ΔMeth $P = 6.0E-04$). Lastly, plots showing the distribution of the median ΔMeth and relative ΔMeth in breast cancers in the **e** hypomethylation direction in CpG sties with low native levels (ΔMeth $P = 0.01$, relative ΔMeth $P = 3.0E-06$) and in the **f** hypermethylation direction in CpG sites with high native levels (ΔMeth $P = 0.003$, relative ΔMeth $P = 2.0E-04$). Asterisk is statistically different (Wilcoxon rank sum test, two-sided $P < 0.001$). NS is non-statistically different (Wilcoxon rank sum test, two-sided $P > 0.001$).

## Identifying biologically important sites

Since the evidence overwhelmingly suggests that DNA methylation alterations occur as a global rate of change relative to the native DNA methylation level, we investigated whether selecting for large relative ΔMeth would enrich for a different and more biologically important set of CpG sites in comparison to ΔMeth. We first combined all the cancer samples into a single dataset and selected the 10% most frequently altered CpG sites in both the hypomethylation and hypermethylation directions (Fisher's Exact $P ≤ 5.25E-78$). We found that the 10% most frequently altered CpG sites had native DNA methylation levels in all quantile intervals.

From the 10% most frequently altered CpG sites, we further selected the top 10% CpG sites with the largest median ΔMeth and relative ΔMeth. In the hypomethylation direction, the largest 10% ΔMeth selected 0 CpG sites with native levels between 0.0–0.25 (Fig. 5a left side). Similarly, in the hypermethylation direction, 0 CpG sites with native levels between 0.75–1.0 were selected (Fig. 5a right side). In both directions, a clear association between the native levels and the degree of ΔMeth was observed (Fig. 5a). Conversely, the largest 10% relative ΔMeth selected CpG sites with native DNA methylation levels in all quantile intervals in both the hypomethylation and hypermethylation directions (Fig. 5b). Overall, we found that only 20% of the hypomethylation CpG sites and 29% of the hypermethylation

CpG sites were selected by both ΔMeth and relative ΔMeth (Fig. 5c, d and Supplementary Data Table 3, 4). Interestingly, while we found that the CpG sites of the largest 10% ΔMeth and relative ΔMeth at different quantile intervals were mostly in OpenSeas in the hypomethylation direction, in the hypermethylation direction, we found the proportions of the genomic regions varied at different quantile intervals (Supplemental Figure 9).

To investigate the biological features of the largest 10% relative ΔMeth and ΔMeth CpG sites, we measured the correlation between DNA methylation and gene expression levels derived from RNA sequencing data of 31 tumor types and 7,910 samples. Genes with statistically significant correlation ($P < 0.001$) were identified in the hypomethylation and hypermethylation directions by ΔMeth and relative ΔMeth. We found that ΔMeth and relative ΔMeth identified a similar number of genes with statistically significant correlation and that the strength of the correlations were also similar (Supplemental Figures 10–13). Although the largest 10% relative ΔMeth selected for CpG sites with native DNA methylation levels in all quantile intervals, no genes with statistically significant correlation were selected at the highest quantile interval in the hypomethylation direction. Overall, only a minority of genes with statistically significant ($P < 0.001$) and strong correlations (rho $≥ |0.2|$) were identified by both ΔMeth and relative ΔMeth in the hypomethylation and hypermethylation

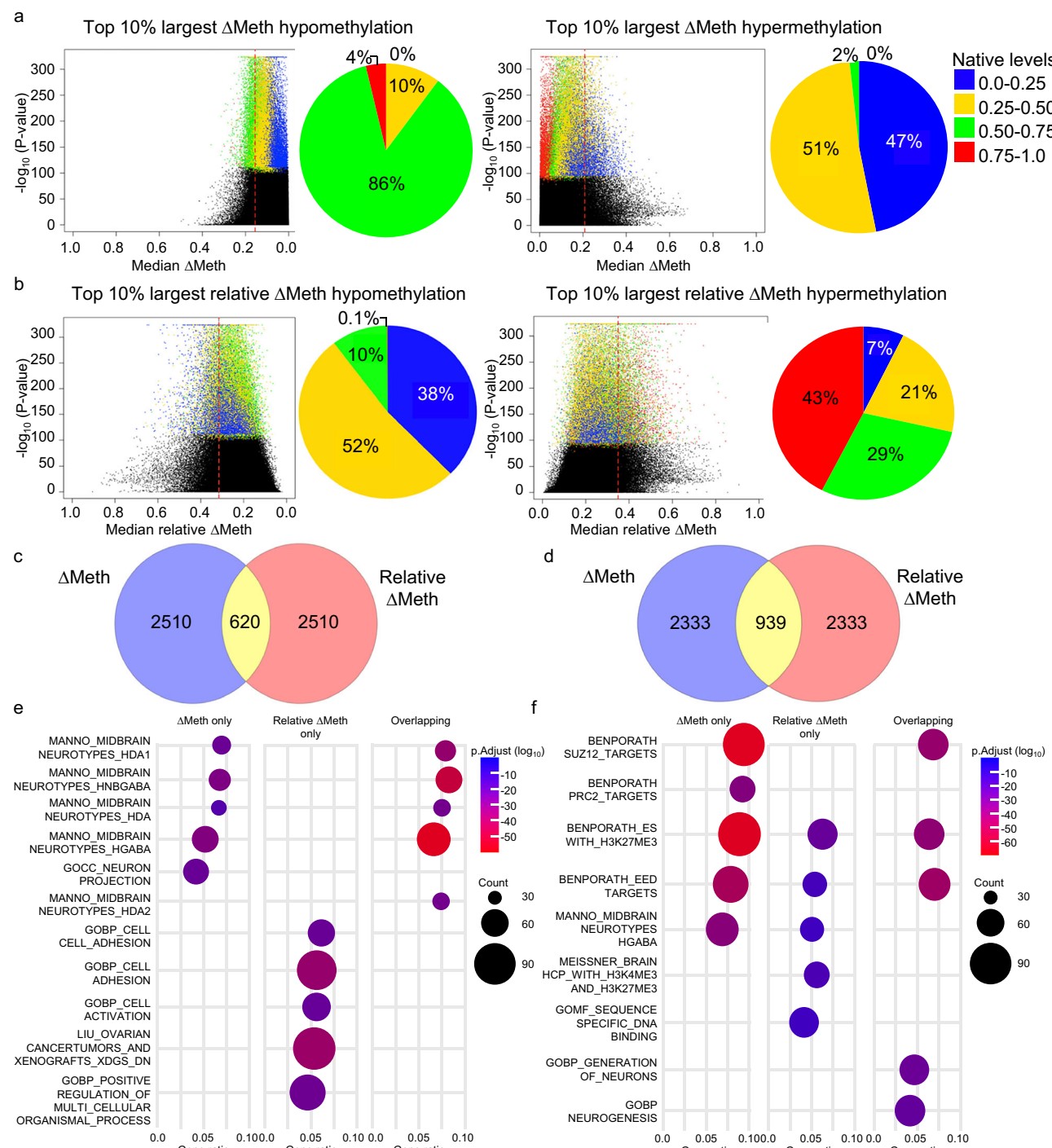

**Fig. 5 | Identification of large alterations using the ΔMeth and relative ΔMeth approach.** Dot plots and corresponding pie charts showing the 10% most frequently altered CpG sites (colored dots) with the 10% largest (red dotted line) as calculated by (**a**) ΔMeth and (**b**) relative ΔMeth. The P values are calculated using the Fisher's Exact test, two-sided. Venn diagrams showing the number of CpG sites identified by ΔMeth and relative ΔMeth in the (**c**) hypomethylation and (**d**) hypermethylation direction. Dot plots showing the top 5 most statistically significant GSEA pathway results from the genes with statistically significant and strong correlation between DNA methylation and expression levels identified in the (**e**) hypomethylation and (**f**) hypermethylation direction. The p.Adjust P value has been adjusted for multiple comparisons.

directions, 25% and 28% respectively (Supplementary Data Tables 5, 6 and Supplemental Figure 11).

From the Gene Set Enrichment Analysis (GSEA), using the genes with statistically significant and strong correlations, we found that the genes uniquely identified by the 10% largest ΔMeth were enriched for pathways important for the midbrain (Fig. 5e and Supplementary Data Table 7, 9). Similar brain and neuron pathways were also identified

when investigating the statistically significant genes uniquely identified in the four most frequent tumor types (breast, prostate, lung and colorectal) independently (Supplemental Figure 14a). Surprisingly, in the combined cancer dataset, the genes that were uniquely identified by the 10% largest relative ΔMeth showed enrichment for multiple pathways with known importance for tumor progression including increased metabolism, cell signaling, immune activation, and cell

adhesion (Fig. 5e and Supplementary Data Table 8)[25–28]. Interestingly, when investigating the relative ΔMeth genes uniquely identified in the four most frequent tumor types independently, we found that the pathways identified using hypomethylation alterations were largely tumor specific, although similar types of pathways, such as cell signaling and immune activation, were found in multiple tumor types (Supplemental Figure 14b). In the hypermethylation direction, we found that the genes identified by both ΔMeth and relative ΔMeth were enriched for pathways important for polycomb regulated stemness (Fig. 5f and Supplementary Data Tables 10–12). Unlike the alterations identified in the hypomethylation direction, in the hypermethylation direction, we found a large degree of shared polycomb-regulated stemness pathways when investigating breast, prostate, lung and colorectal cancers independently (Supplemental Figure 14c, d).

### Investigating and comparing Δ*M*-value hypomethylation alterations

Since the relative ΔMeth approach identified more biologically significant hypomethylation alterations in comparison to ΔMeth, we investigated whether other approaches that can select for hypomethylation alterations at CpG sites that have low native levels (a distinct feature of the relative ΔMeth approach) can also be used to enrich for genes in pathways important for tumor development. To accomplish this, we used the well-established Δ*M*-value approach[29], a method that increases the resolution of DNA methylation differences as the DNA methylation levels near 0.0 and 1.0. We compared the top 10% largest hypomethylation alterations of the Δ*M*-value to the top 10% largest hypomethylation alterations of relative ΔMeth. As expected, we found that the largest Δ*M*-value selected CpG sites with native DNA methylation levels in all quantile intervals (Fig. 6a and Supplemental Figure 15).

After selecting for genes with statistically significant and strong correlation between gene expression and DNA methylation levels, we found that there were more overlapping genes identified between the largest *M*-value and relative ΔMeth (35%) in comparison to the overlapping genes between the largest ΔMeth and relative ΔMeth (25%) (Fig. 6b and Supplemental Figure 11c). However, similar to ΔMeth, from the GSEA, we found that the genes identified using the Δ*M*-value approach were in pathways important for the midbrain (Fig. 6c and Supplementary Data Tables 13, 15). Once again, the genes uniquely identified by relative ΔMeth were enriched for pathways with known biological importance to cancer development (Fig. 6c and Supplementary Data Table 14). Thus, we conclude that the ability of relative ΔMeth to enrich for biologically significant alterations in the hypomethylation direction is not simply due to the selection of CpG sites that have low native levels.

### Hypomethylation alterations in distinct genomic compartments

Since the loss of DNA methylation at different genomic compartments has been shown to have distinct biological consequences for tumor development[30–33], we investigated whether large, uniquely identified, relative ΔMeth hypomethylation alterations that had statistically significant and strong correlation with gene expression, occur in different genomic compartments compared to large ΔMeth alterations. Since the Illumina 450 K DNA methylation array does not provide the genomic coverage required to accurately call different genomic compartments, we first downloaded the genomic positions of previously identified genomic compartments, partially methylated domains (PMDs), highly methylated domains (HMDs), and genomic domains that are neither PMDs nor HMDs (PMDs/HMDs), which were found to be common in multiple tissue types[33]. Since the genomic domains can be tissue specific, we identified the array CpG sites that showed concordance between the DNA methylation levels of normal-adjacent tissues and the previously identified genomic compartments. Because of their frequency and the availability of normal-adjacent

tissues, we focused our investigation on breast cancer and colorectal cancer. We defined the CpG sites of the normal-adjacent tissues to be concordant with the previously identified genomic compartments if the median DNA methylation level was >60% in common HMDs, between 30% and 60% in PMDs, and <30% in neither PMDs/HMDs.

After we selected the top 10% largest ΔMeth and relative ΔMeth alterations from the CpG sites that had concordance with the genomic compartments, we found that uniquely identified alterations by ΔMeth occurred most frequently in common HMDs in both breast and colorectal cancer tissues (Fig. 7a, b). We also found that the unique relative ΔMeth alterations occurred more frequently in common PMDs and in neither PMDs/HMDs in comparison to ΔMeth. Next, we investigated whether the alterations in the three genomic compartments enriched for genes in different biological pathways. Similar to our previous findings, the genes uniquely identified by the top 10% largest ΔMeth in both breast and colorectal cancers were largely enriched for pathways which are confounding for tumor development (Fig. 7c, e and Supplementary Data Tables 16, 19). Interestingly, we found that the unique genes in common PMDs identified in the top 10% largest relative ΔMeth in both breast and colorectal cancers were enriched for both pathways important for cancer development and brain development, while the genes in both common HMDs and in neither PMD/HMDs were largely enriched in pathways known to be important for tumor development (Fig. 7d, f and Supplementary Data Tables 17, 18, 20–22). Interestingly, in both breast and colorectal cancers, we found that alterations in common HMDs were largely enriched for genes in immune cell pathways. In conclusion, our data supports the findings that DNA methylation alterations in different genomic compartments have different biological roles for tumor development and that these differences are more obvious when using the relative ΔMeth approach in comparison to the ΔMeth approach.

## Discussion

The goal of this study was to determine whether DNA methylation alterations occur as an absolute level change or as a global rate of change relative to the native level. If alterations generally occur as a global rate of change relative to the native level, we also wanted to determine whether selecting for CpG sites that have large relative ΔMeth can enrich for alterations that have more biological significance for cancer progression in comparison to those selected using the ΔMeth calculation. Here, we provide multiple lines of evidence supporting the idea that DNA methylation alterations occur as a global rate of change relative to the native level. This evidence includes both the overlapping hypomethylation and hypermethylation distribution profiles and similar rates of change between relative ΔMeth and the relative frequency of the alteration at CpG sites with varying native levels. Furthermore, we also found that the relative ΔMeth can more uniquely detect cancer signatures that associate with age, smoking and H-CIMP in comparison to ΔMeth. These findings provide new insights into the nature in which alterations occur: as a global rate of change relative to the native level.

Next, we investigated whether large relative ΔMeth enriches for different and more biologically important alterations in comparison to ΔMeth. We found that very few CpG sites were selected by both large ΔMeth and relative ΔMeth, which suggests that ΔMeth has likely heavily skewed which CpG sites have been previously selected to be further investigated. In the hypermethylation direction, we found that large ΔMeth and relative ΔMeth enriched for genes in pathways important for polycomb-regulated stemness. This provides credence for the usage of relative ΔMeth as there is a wealth of knowledge regarding the biological importance of hypermethylation events in polycomb-regulated stem cell genes[34,35].

In the hypomethylation direction, we found that the genes identified only by ΔMeth were enriched in brain pathways, a confounding pathway for cancer development, which is a common result for

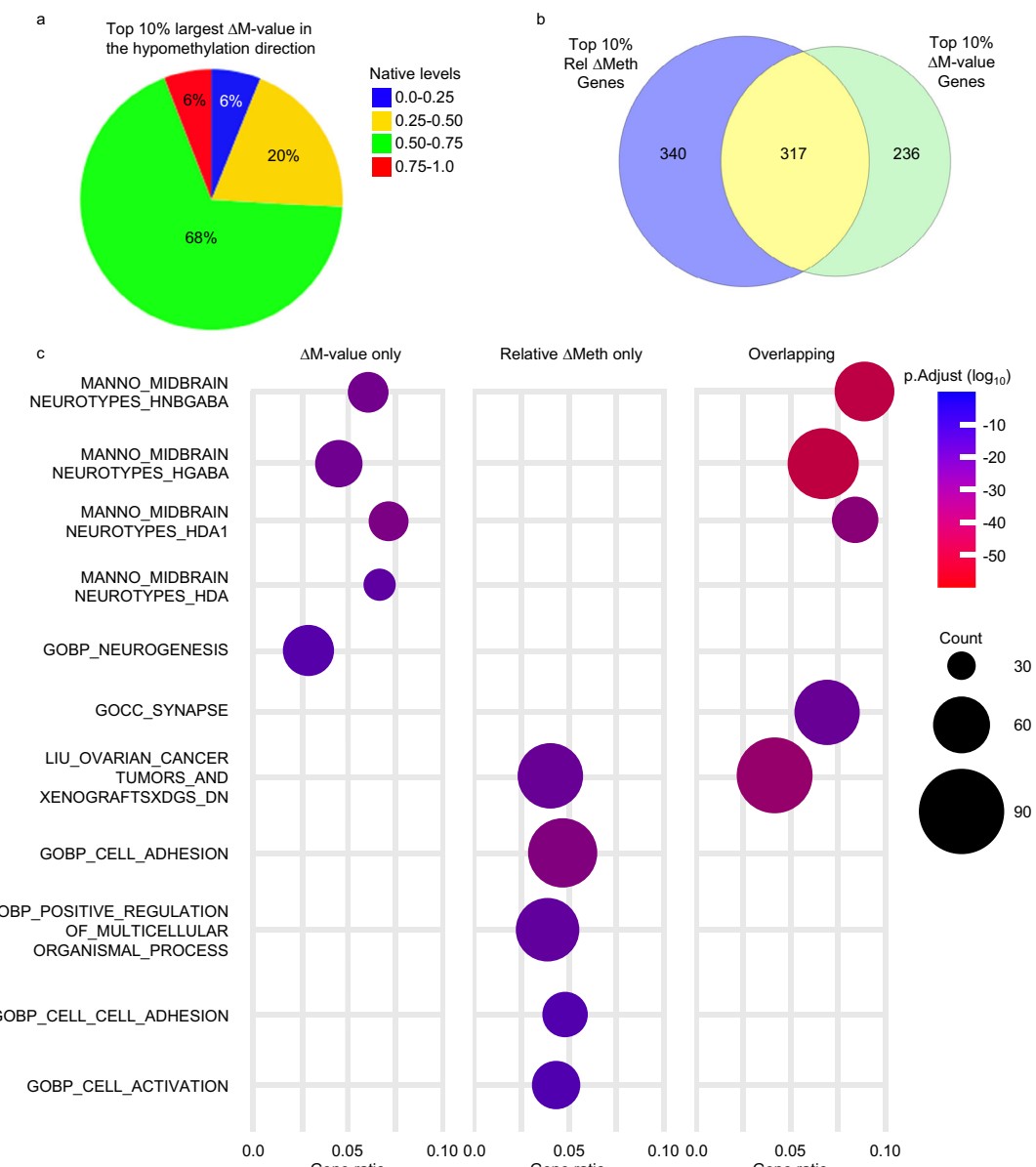

**Fig. 6 | Identification and comparison of large hypomethylation alterations using the ΔM-value and relative ΔMeth approach. a** Pie chart showing the percentages of ΔM-value selected CpG sites at different native DNA methylation level quantile intervals. **b** Venn diagram showing the number of genes that have significant and strong correlation between DNA methylation and RNA expression selected by ΔM-value and relative ΔMeth. **c** Dot plots showing the top 5 most statistically significant GSEA pathway results from the genes with statistically significant and strong correlation between DNA methylation and expression levels identified by ΔM-value and relative ΔMeth. The *p*.Adjust *P* value has been adjusted for multiple comparisons.

hypomethylation events[13]. These hypomethylation events are commonly explained as alterations that increase genomic instability and may lead to the reactivation of transposable elements. Conversely, the genes identified only by relative ΔMeth were enriched in multiple pathways known to be important for cancer development including pathways for increased metabolism, cell signaling, immune activation, and cell adhesion (a cellular function known to be important for tumor metastasis[25–28]).

One significant difference between the ΔMeth and relative ΔMeth approach is that the relative ΔMeth selects for more hypomethylation alterations at CpG sites that have low native levels. This raises the question whether other approaches that select for CpG sites with low native levels, such as the ΔM-value, would also enrich for biologically important hypomethylation alterations. While we found that both the

relative ΔMeth and ΔM-value approaches identified hypomethylation alterations at CpG sites that have low native levels, similar to the ΔMeth approach, we found that the genes identified only by ΔM-value were enriched in pathways important for the brain. Conversely, we found that genes identified uniquely by relative ΔMeth were enriched in pathways important for tumor development. This result suggests that the relative ΔMeth ability to enrich for biologically significant hypomethylation alterations is not simply due to the selection of CpG sites that have low native levels. This finding strengthens the argument that using an approach that mirrors how DNA methylation alterations occur, a change relative to the native level, can increase the ability to enrich for more biologically significant alterations.

Next, we investigated whether the hypomethylation alterations identified using the relative ΔMeth and ΔMeth approaches associated

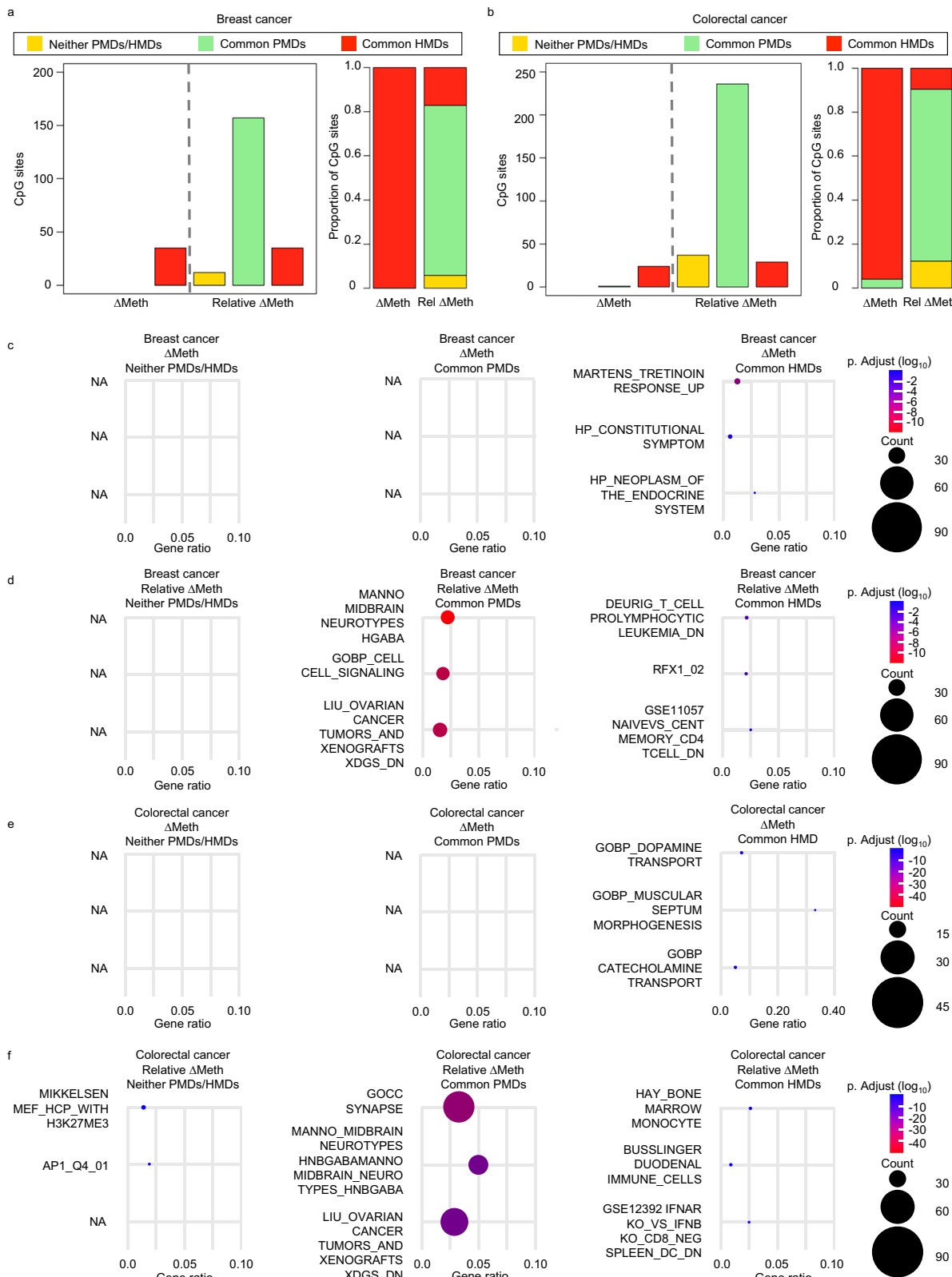

**Fig. 7 | Characteristics of the largest unique hypomethylation alterations at different genomic compartments.** Bar plots showing the number and proportion of CpG sites that have strong correlation between DNA methylation and gene expression in the genomic compartments of PMDs, HMDs and neither PMDs/HMDs in (**a**) breast cancer and (**b**) colorectal cancer. Dot plots showing the top 3 most statistically significant GSEA pathway results from the genes with statistically significant and strong correlation between DNA methylation and expression levels at different genomic compartments identified by **c** ΔMeth and (**d**) relative ΔMeth in breast cancer and (**e**) ΔMeth and (**f**) relative ΔMeth in colorectal cancer. The *p*.Adjust is the *P* value adjusted for multiple comparisons.

at different frequences within the genomic compartments of PMDs, HMDs and neither PMDs/HMDs, which studies have shown have distinct biological functions for tumor development[30–33]. After we found that the large relative ΔMeth alterations that had strong correlation with gene expression occurred more frequently in common HMDs and neither PMDs/HMDs in comparison to large ΔMeth alterations, we used GSEA to investigate the biological significances of these alterations. We found that the genes identified by large ΔMeth were only in HMDs and were largely enriched in pathways that are confounding for tumor development. Conversely, we found that the relative ΔMeth alterations in all genomic compartments were enriched in genes in pathways known to be important for tumor development. Interestingly, we found that different genomic compartments had distinct importance for tumor development. For example, in both breast and colorectal cancers, alterations in genes in HMDs were largely enriched for immune pathways. These findings provide a biological explanation for why the relative ΔMeth approach enriches more biologically significant alterations in comparison to ΔMeth and suggests that biological features can be better defined with the relative ΔMeth approach in comparison to the ΔMeth approach.

Although both large ΔMeth and relative ΔMeth hypomethylation events may be important, the pathways identified using large relative ΔMeth are more obvious for tumor progression in comparison to those identified by large ΔMeth. Thus, we conclude that the unbiased selection of large relative ΔMeth can identify more biologically important alterations in comparison to ΔMeth. Furthermore, our findings also suggest that the usage of the ΔMeth calculation has likely heavily skewed our understanding of DNA methylation biology and that the largest and, likely most important, DNA methylation alterations in the cancer epigenome remain undefined.

In conclusion, we present evidence that DNA methylation alterations largely occur as a change relative to the native level and that the relative ΔMeth calculation provides an unbiased approach to identify new classes of biologically important DNA methylation alterations. Although we have shown the benefits of the relative ΔMeth approach to measure and compare DNA methylation alterations, how to best incorporate this approach into common methodologies used to investigate DNA methylation level changes need to be further investigated. We are hopeful that using relative ΔMeth to measure DNA methylation alterations will be the beginning of a new era of DNA methylation analysis.

## Methods

### Cell lines

For the cell line DNA methylation data, which were measured using the Illumina Human Methylation 450 K array, the probe-specific native DNA methylation levels were derived from the levels of the untreated cells. We compared the cell lines 3BKO and 3ABDKO before and after 5 days of 5-azacytidine treatment. We considered the DNA methylation level to be changed if the ΔMeth was ≥0.01. For probes with DNA methylation level changes in the hypomethylation direction, the ΔMeth (untreated cell DNA methylation level– treated cell DNA methylation level) and relative ΔMeth (ΔMeth/native DNA methylated level) were calculated. The R function density was used to make the density plots.

### Sample selection

To determine the upper and lower probe-specific native DNA methylation levels for the normal tissues, we used the OutlierMeth package (https://github.com/bdowns4/OutlierMeth) at the significance levels P0.01 using all of the available normal tissues. A detailed description of sample acquisition and usage of the OutlierMeth package can be found in our previously published manuscript[21]. In short, all normal tissues available in the OutlierMeth package were included in this study (25 normal tissue types N = 1991). All TCGA cancer DNA methylation data used in this study was derived using the Illumina Human Methylation 450 K array and was comprised of 31 tumor types and 8551 samples. All paired TCGA cancer RNA-Seq and Illumina Human Methylation 450 K array data was used and was comprised of 31 tumor types and 7910 samples.

### Calculating ΔMeth, relative ΔMeth, and ΔM-value for tumor tissues

We considered the cancer tissue DNA methylation level to be aberrant if it was less than the lower native DNA methylation level or greater than the upper native DNA methylation level. We used the equation (|aberrant cancer DNA methylation level−native DNA methylation level|) to calculate the ΔMeth in the hypomethylation and hypermethylation direction. We used the equations (ΔMeth/native DNA methylation level) and (ΔMeth/(native DNA unmethylated level)) to calculate the relative ΔMeth in the hypomethylation and hypermethylation direction respectively. The OutlierMeth package function deltMeth and relMeth was used to calculate all ΔMeth and relative ΔMeth. The M-value was calculated using the equation $\log_2(Beta_i/1\text{-}Beta_i)$ and the hypomethylation ΔM-value was calculated using the equation ($M\text{-value}_{native} - M\text{-value}_{cancer}$).

### Distribution profiles of ΔMeth and relative ΔMeth

To ensure that each tumor type carried equal weight, we analyzed the tumor types separately before the results were combined. To limit the effect that the relative frequency of the alteration (number of samples with an aberrant DNA methylation level$_i$/total number of samples$_i$) has on the distribution profiles of ΔMeth and relative ΔMeth, the R function aggregate was used to group the CpG sites based on the relative frequency of the aberrant events and to calculate the median value for each group (aggregated median). The R function density was used to make the density plots. The R function Wilcox.test (two-sided) was used to compare the distributions of the ΔMeth and relative ΔMeth levels. All distributions of the ΔMeth and relative ΔMeth levels were found to be statistically different (Wilcoxon rank sum test $P < 0.001$) unless described as not statistically different (NS) (Wilcoxon rank sum test $P > 0.001$). The R function sample was used to randomly generate DNA methylation levels for the random DNA methylation level dataset. The R package pheatmap was used to generate the heatmaps[36].

### The rate between the ΔMeth and relative ΔMeth and the relative frequency of the alteration

The aggregated median ΔMeth and relative ΔMeth were calculated, as described above, separately for the 31 tumor types before the results were combined. The R function hist was used to generate the histograms showing the frequency of the data based on the relative frequency of the alteration. The R functions lm was used to calculate the slopes and the function coef was used to calculate the coefficient $P$ values. All slopes were found to be statistically different in comparison to each other (z-statistic $P < 0.001$) unless described as not statistically different (NS) (z-statistic $P > 0.001$). To calculate the z-statistic we used the following equation $z = (Coeff_1 - Coeff_2)/\sqrt{(Se_1^2 - Se_2^2)}$[37]. The R function pnorm was used to calculate the $P$ value using the z-statistic.

### Cancer signatures

The R function sample was used to randomly select 1000 cancer samples from the combined TCGA dataset. We also used the R function sample to randomly select the CpG sites that were altered for the generated DNA methylation signatures. In total, we generated 8 test conditions by altering a range of CpG sites (10, 20, 30, or 40%) by a percentage (10, 20, 30, or 40%) of gain or loss of DNA methylation relative to the cancer samples' native level. We generated $n = 10$ DNA methylation signatures for each of the test conditions. The Wilcoxon

rank sum test was used to determine whether the aggregated median ΔMeth and median relative ΔMeth values were statistically different ($P < 0.001$) compared to the native levels of the selected 1000 TCGA cancer samples.

Colorectal tumors (COADREAD) from TCGA were considered to be high-CIMP (H-CIMP) if they carried a BRAF mutation or were determined H-CIMP in the TCGA manuscript[24]; ($n = 126$). Non-H-CIMP tumors did not have a BRAF mutation and were not previously determined as H-CIMP; ($n = 503$). Lung cancer patients (LUAD) from TCGA were considered a smoker if they had >5 pack years. The sample size for LUAD smokers $n = 343$ and non-smokers $n = 166$. Breast cancer patients from TCGA (BRCA) were divided into two groups, ≤50; ($n = 331$) and >50 age; ($n = 766$) at primary diagnosis.

### The top 10% most altered CpG sites identified by ΔMeth and relative ΔMeth

The R function quantile was used to select the top 10% most frequently altered DNA methylation CpG sites and the CpG sites with the 10% largest ΔMeth and relative ΔMeth. The R function Fisher test (two-sided) was used to calculate the $P$ values for the altered CpG sites between the compiled TCGA cancer dataset ($n = 8551$) and the normal tissue dataset ($n = 1991$). The package eulerr was used to make the Venn diagrams. The R function cor was used to calculate the rho correlation values and spearman coefficient $P$ values. The Gene Set Enrichment Analysis (GSEA) was performed on the GSEA website (https://www.gsea-msigdb.org). The per-gene methylation information was derived from the 450k annotation file downloaded from the Illumina website. The common PMDs, HMDs and neither PMDs/HMDs genomic locations[33] were downloaded from the UCSC Genome Browser website. The CpG sites of the 450 K array from the normal-adjacent tissue was defined as concordant with the genomic compartment if the median DNA methylation level was > 60% in common HMDs, between 30% and 60% in PMDs, and <30% in neither PMDs/HMDs.

### Statistics and reproducibility

No statistical method was used to predetermine sample size. No data were excluded from the analyses.

The R function sample was used to randomly select the CpG sites that were altered for the generated DNA methylation signatures. The experiments analyzing data from cancer tissues were not randomized. The investigators were not blinded to allocation during experiments and outcome assessment.

### Reporting summary

Further information on research design is available in the Nature Portfolio Reporting Summary linked to this article.

## Data availability

3BKO and 3ABDKO cell line data was downloaded from the Gene Expression Omnibus (GEO) database (GSE51815. and GSE68344). The TCGA tumor DNA methylation data, mutational data, RNA sequencing data and clinical data were downloaded from the Firebrowse site (http://firebrowse.org).

## Code availability

The OutlierMeth package is on GitHub (https://github.com/bdowns4/OutlierMeth)[38] and the custom code used for the analyses in this study is available upon reasonable request. The R version 4.2.3 was used for the analysis in this manuscript.

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

## Acknowledgements

The authors would like to thank Dr. Claudia Mercado Rodriguez, Dr. Harharan Easwaran, Dr. Leslie Cope, and Dr. Christopher Umbricht for their generous advice for the preparation of this manuscript. We would also like to thank all of the students of the Tza-Huei Wang lab for their support.

## Author contributions

B.D. conceived the idea for this work. B.D. completed the bioinformatic analysis. J.H., J.P., H.L., T.W., T.R.P., K.H., and T.H. wrote, designed and contributed to the interpretation of the data and the preparation of the manuscript.

## Competing interests

The authors declare no competing interests.
