## [Transparent Peer Review file · Nature Communications]

An unbiased approach to measure aberrant DNA methylation alterations

Corresponding Author: Dr Bradley Downs

Version 0:

Reviewer comments:

Reviewer #1

(Remarks to the Author)

In this manuscript, Bradley and colleagues propose the use of relative delta beta values as an alternative to absolute difference in delta beta values as a metric to measure differences between CpG methylation levels in aberrant vs. native conditions. The main working hypothesis supporting the use of relative delta beta values is that DNA methylation alterations occur as global rate of change rather than an absolute change relative to the native level and this latter aspect could potentially lead to measurement biases when using delta beta values in sites characterized by extremely low or high native beta values.

While the adoption of a relative measure could certainly be helpful, especially for signature and biomarkers identification, it should be used with caution, especially when interpreting the biology of the phenomenon under study. In particular, the authors, by comparing results obtained using relative delta beta values with those obtained with delta beta, report to have provided evidence that methylation occur as a global rate of change relative to native levels, but do not provide a biological model or experimental evidence explaining how and in which conditions this can happen: contrary to gene expression, CpG methylation is a binary phenomenon in a cell (i.e. a given site can be methylated or un-methylated), thus, under this terms, working with relative rate of change would imply a model where methylation in a particular site can be found altered in a cell depending on the methylation state in other cells, that could certainly be true in some cases (e.g. under selective pressure) but not in general.

The manuscript needs to be revised to improve its readability as many key points are difficult to interpret. Also, the methods section needs to be improved as it lacks the necessary level of detail in some parts.

Please find below a detailed list of comments.

Major points:

1. Lines 67-68: "relative Δ Meth would be an unbiased approach to enrich for more biologically important alterations in comparison to those identified using Δ Meth": this is questionable, one would expect that at the cell population and phenotypic level a 2 fold relative change in a locus from 0.4 to 0.8 would determine an overall effect remarkably different the one linked to a 2 fold relative change from 0,04 to 0,08.
2. Lines 101-103: "we subset the CpG sites of the Illumina Human Methylation 450K array datasets (3BKO; GSE51815 and 3ABDKO; GSE68344) into four even quantiles based on the untreated cells' DNA methylation level": probes on the Infinium array are not independent and not uniformly distributed along the genome, hence beta values are not independent each other. Did the authors check for unbalance with respect to genomic regions distribution in the defined quantiles intervals?
3. Line 166: The "relative frequency of alteration" should be introduced in the results and its exact calculation should be detailed in the methods.
4. Lines 248-250: The text reports that a similar number of genes with statistically significant association for the absolute and relative delta Meth has been found, while in Figure S9 Panel C there is no plot for the association between relative delta methylation and gene expression levels for native levels 0.75-1.0" in the hypomethylation direction. Please comment this aspect.
5. Lines 358-361: "The R functions lm was used to calculate the slopes and the function coef was used to calculate the coefficient P-values. All slopes were found to be statistically different in comparison to each other (z-statistic $P < 0.001$) unless described as not statistically different (NS) (z-statistic $P > 0.001$)". Please provide details on the statistical test used to compute significance of linear models' coefficients and to compare the slopes.
6. Lines 365-366: "We generated $n=10$ DNA methylation signatures for each test condition.". In the result section and in Supplementary figures S7-S8 only 8 signatures are reported (4 gain and 4 loss) for

each condition. Could the authors please make clear where and how these 10 signatures are used?

7. Supplementary figures S7-S8 legend: Please explicitly state the comparison for the Wilcoxon test.

8. Code availability: Code availability: The presented work is entirely based on in silico analyses and the code used for the analyses is not of marginal importance. It would be important to provide direct access to the code that was used to generate the results. Is there any particular reason why the authors have decided to release the code only upon reasonable request?

Minor points:

1. Lines 103, 128: The usage of the term quantile to indicate intervals is confusing, I suggest using the term quantile intervals.

2. Figure 3 Legend: "Histograms show the frequency of data." Please provide a more detailed description for the reported frequencies.

3. Supplementary Figure S6: The association between Delta Meth and relative frequency of alteration is not present in all tumors, in particular for the native levels 0.75-1.0 there is one tumor for which the association has the opposite direction. Is there any particular reason for this?

4. Lines 335-336: "We considered the cancer tissue DNA methylation level to be aberrant if it fell outside of the native DNA methylation level." Please provide a more specific definition.

5. Lines 336-337: "We used the equation (aberrant cancer DNA methylation level – native DNA unmethylated level) and (aberrant cancer DNA methylation level – native DNA methylation level) to calculate the Δ Meth in the hypomethylation and hypermethylation direction respectively". Shouldn't it be the opposite for the hypomethylation direction (i.e. native DNA unmethylated level - aberrant cancer DNA methylation level) ?

6. Lines 346-347: Please provide more details on the aggregation of sites based on the relative frequency of aberrant events and, in particular, on how the grouping variable is defined.

7. Supplementary figures S7-S8 legend: "in the hypomethylation direction a measured". Typo here.

Reviewer #2

(Remarks to the Author)

Downs et al. propose a new metric for quantifying changes in DNA methylation. Instead of the standard Δ Meth (tumour – normal, or treatment – control), they introduce relative Δ Meth, which divides Δ Meth by the baseline methylated level when the change is a loss (hypomethylation) or by the baseline unmethylated level when the change is a gain (hypermethylation). Using relative Δ Meth, the authors show that methylation changes appear more consistent across CpGs with different starting values, whereas absolute Δ Meth is strongly influenced by the native level. They argue that relative Δ Meth is a less biased way to detect biologically meaningful alterations. Gene-set enrichment analysis (GSEA) of the CpGs with the highest relative Δ Meth in TCGA tumours highlights cancer-specific pathways, whereas the same analysis based on absolute Δ Meth points to broader, less disease-specific pathways.

The core idea of treating the mean relative Δ Meth as background and identifying loci whose relative Δ Meth exceeds that background is potentially important.

However, we believe several key issues need to be addressed (details below).

1 Global versus regional change

A central assertion—"DNA-methylation alterations occur as a global rate of change relative to the native level"—is problematic. Although the net gain or loss is, on average, proportional to the starting level, the apparent consistency arises from averaging highly heterogeneous effects with distinct biological implications. Numerous studies have shown that DNA-methylation changes cluster in specific compartments: e.g. in cancer, focal island hypermethylation and large-scale hypomethylation tend to occur in late-replicating, lamina-associated regions (also known as PMDs; see doi: 10.1038/nsmb.2518, 10.1038/nrm.2015.31, 10.1038/ng.969). Hypomethylation of PMDs, often age-associated, also contributes to tumorigenesis (doi: 10.1038/s41588-018-0073-4). Averaging all Open-Sea probes therefore yields a figure that does not represent either highly methylated domains (HMDs) or PMDs.

Indeed, the first experiment presented, 5-AZA treatment, may be the only one that produces a true genome-wide loss: under 5-AZA, virtually every Open-Sea CpG loses methylation by a similar relative Δ Meth, whereas in cancer only the Open-Sea CpGs within PMDs lose methylation.

Specific recommendations

a) Acknowledge prior evidence that methylation loss is compartment-specific, not uniform.

b) Re-evaluate Δ Meth and relative Δ Meth separately within late-replicating / lamina-associated domains (or PMDs / hypomethylated blocks) to test whether the new metric captures biologically meaningful changes there.

2 Biological relevance of the selected CpGs

The authors claim that relative Δ Meth identifies more relevant CpGs, based on GSEA of the top 10 % most variable sites. This apparent improvement likely reflects the inclusion of low-baseline hypomethylated CpGs and high-baseline hypermethylated CpGs. Demonstrating functional importance requires further evidence, e.g. by correlating methylation and expression specifically for this subset.

3 Comparison with existing alternatives

An established alternative to Δ Meth is Δ M, where $M = \text{logit}(\beta)$. Is relative Δ Meth superior to Δ M for highlighting biologically meaningful sites? A direct comparison would strengthen the paper.

4 Minor points

a) Baseline definition. The "native" baselines appear to pool all normal tissues. Because DNA methylation is strongly tissue-specific, this will preferentially select the small subset of CpGs that are constitutively high or low in every tissue. Using

matched normal tissue would be more appropriate.

b) CpG-to-gene mapping. For DNAm-expression correlations, how is per-gene methylation derived? Promoter methylation has very different effects from gene-body methylation; the analysis should stratify by genomic context.

Version 1:

Reviewer comments:

Reviewer #1

(Remarks to the Author)

The authors have taken action to address all the concerns raised by this reviewer.

The text is now clearer, and the previously missing details about various aspects have been sufficiently described.

Reviewer #2

(Remarks to the Author)

The authors have partially addressed my concerns from the original review. In particular, I appreciate the work done in response to the issue of methylation patterns (both normal/baseline and 'aberrant') being very different between different genomic regions. My remaining concern is that the new approach of studying changes in genomic regions defined as HMDs vs PMDs for all cancers grouped together is difficult to interpret since genomic methylation patterns are so tissue specific. (In fact a list of HMDs and PMDs for a given sample is a great predictor of which tissue type the sample comes from). In my opinion, this analysis would have to be done in a tissue-specific manner (or at least a subset of individual tissues) to be meaningful. (I.e. define HMDs, PMDs for that tissue, and then relate it to methylation alterations in the same tissue).

Version 2:

Reviewer comments:

Reviewer #2

(Remarks to the Author)

I appreciate this additional breast and colon tissue analysis which addresses my earlier concern.

One final comment to the authors: For future work (i.e. in my opinion not necessary for this manuscript) it could be interesting to understand what happens specifically in regions that are discordant between tissues (e.g. regions that are HMD in breast and PMD in colon).

RESPONSE TO REVIEWERS' COMMENTS

Reviewer #1 (Remarks to the Author):

In this manuscript, Bradley and colleagues propose the use of relative delta beta values as an alternative to absolute difference in delta beta values as a metric to measure differences between CpG methylation levels in aberrant vs. native conditions. The main working hypothesis supporting the use of relative delta beta values is that DNA methylation alterations occur as global rate of change rather than an absolute change relative to the native level and this latter aspect could potentially lead to measurement biases when using delta beta values in sites characterized by extremely low or high native beta values.

While the adoption of a relative measure could certainly be helpful, especially for signature and biomarkers identification, it should be used with caution, especially when interpreting the biology of the phenomenon under study.

In particular, the authors, by comparing results obtained using relative delta beta values with those obtained with delta beta, report to have provided evidence that methylation occur as a global rate of change relative to native levels, but do not provide a biological model or experimental evidence explaining how and in which conditions this can happen: contrary to gene expression, CpG methylation is a binary phenomenon in a cell (i.e. a given site can be methylated or un-methylated), thus, under this terms, working with relative rate of change would imply a model where methylation in a particular site can be found altered in a cell depending on the methylation state in other cells, that could certainly be true in some cases (e.g. under selective pressure) but not in general.

The manuscript needs to be revised to improve its readability as many key points are difficult to interpret. Also, the methods section needs to be improved as it lacks the necessary level of detail in some parts.

We would like to thank Reviewer 1 for all of their comments and suggestions. In our resubmitted manuscript, we made many changes throughout our article to improve the readability and have included new Supplemental Figure S11-13 to more clearly explain our findings. We believe that these modifications have improved the clarity of our manuscript significantly.

Please find below a detailed list of comments.

Major points:

1. Lines 67-68: “relative Δ Meth would be an unbiased approach to enrich for more biologically important alterations in comparison to those identified using Δ Meth”: this is questionable, one would expect that at the cell population and phenotypic level a 2 fold relative change in a locus from 0.4 to 0.8 would determine an overall effect remarkably different the one linked to a 2 fold relative change from 0,04 to 0,08.

We agree with the reviewer that there is likely a point at which the absolute level change would be biologically informative. Since the degree of DNA methylation level change does not provide information regarding whether, and at which point, gene expression level undergoes a biologically significant change, we focused on the genes that showed a statistically significant correlation between expression levels and DNA methylation levels.

In an attempt to understand whether the scenario described above is having a large effect on the ability of relative Δ Meth to enrich for important alterations, we included new Supplemental Figures S11-13, showing the number of genes and the levels of correlation between gene expression and DNA methylation at CpG sites at different quantile intervals.

Overall, we found that the relative Δ Meth was identifying a similar number of genes that had similar strengths of correlation between expression levels and DNA methylation levels in comparison to Δ Meth. Thus, we conclude that the scenario described above is rare and is having a relatively small effect on the ability of relative Δ Meth to enrich for important alterations.

2. Lines 101-103: “we subset the CpG sites of the Illumina Human Methylation 450K array datasets (3BKO; GSE51815 and 3ABDKO; GSE68344) into four even quantiles based on the untreated cells’ DNA methylation level”: probes on the Infinium array are not independent and not uniformly distributed along the genome, hence

beta values are not independent each other. Did the authors check for unbalance with respect to genomic regions distribution in the defined quantiles intervals?

We would like to thank the reviewer for raising this important point. We have investigated the distributions of CpG native levels across the genomes of both cell lines. While we found that the probes were not uniformly distributed across the chromosomes, we did not find noticeable levels of unbalance with respect to the genomic regions and the native levels in either 3BKO or 3ABDKO cell lines. Plots of the CpG native levels across the whole genome and chromosome 1 of cell lines 3BKO and 3ABDKO are shown below.

3. Line 166: The “relative frequency of alteration” should be introduced in the results and its exact calculation should be detailed in the methods.

We apologize for the lack of clarity and we have included the calculation for the relative frequency of alterations in the Results and Methods sections. (Lines 134-135 and 413-414).

4. Lines 248-250: The text reports that a similar number of genes with statistically significant association for the absolute and relative delta Meth has been found, while in Figure S9 Panel C there is no plot for the association between relative delta methylation and gene expression levels for native levels 0.75-1.0” in the hypomethylation direction. Please comment this aspect

We would like to apologize for the lack of clarity regarding this point. We have included a new Supplemental Figure S11 to show more clearly the number of genes with statistically significant and strong correlation between expression level and DNA methylation level.

The reviewer is correct in that no genes were selected from CpG sites with native levels 0.75-1.0 in Figure S9 Panel C (now called Figure S10 Panel C). Although the relative Δ Meth selected CpG sites in all quantile intervals, very few (0.1% of the selected CpG sites; Figure 5B) were from the top quantile interval in the hypomethylation direction. We found that none of these CpG sites correlated with the expression level of their corresponding gene, thus, there is no data in the plot for Figure S10C. We have included a comment regarding this point in the Results section. (Lines 250-252).

5. Lines 358-361: “The R functions lm was used to calculate the slopes and the function coef was used to calculate the coefficient P-values. All slopes were found to be statistically different in comparison to each other (z-statistic $P < 0.001$) unless described as not statistically different (NS) (z-statistic $P > 0.001$)”. Please provide

details on the statistical test used to compute significance of linear models' coefficients and to compare the slopes.

We apologize for the lack of details and we have provided the calculation used for this statistical test in the Methods section. (Lines 430-432).

6. Lines 365-366: "We generated n=10 DNA methylation signatures for each test condition.". In the result section and in Supplementary figures S7-S8 only 8 signatures are reported (4 gain and 4 loss) for each condition. Could the authors please make clear where and how these 10 signatures are used?

We would like to apologize for the lack of clarity regarding this experiment. We generated 10 DNA methylation signatures for the four levels of alterations and for the four percentages of CpG sites that were altered. Thus, each condition (box) in the plots of Supplementary Figures S7-S8 has 10 dots. We have provided more details regarding this experiment in the Methods section. (Lines 436-439).

7. Supplementary figures S7-S8 legend: Please explicitly state the comparison for the Wilcoxon test.

We apologize for this oversight. We have stated the comparison for the Wilcoxon test in the legend of Supplementary Figures S7 and S8.

8. Code availability: Code availability: The presented work its entirely based on in silico analyses and the code used for the analyses is not of marginal importance. It would be important to provide direct access to the code that was used to generate the results. Is there any particular reason why the authors have decided to release the code only upon reasonable request?

We would like to thank the reviewer for raising this point. While this study is based on in silico analyses, besides the relative Δ Meth calculation, which is based on a previously published package and the code is provided, the majority of this study uses relatively simple code, such as calculating median values and sorting and selecting data. Although unlikely, for transparency purposes, we decided to offer the code if requested.

Minor points:

1. Lines 103, 128: The usage of the term quantile to indicate intervals is confusing, I suggest using the term quantile intervals.

We apologize for the confusion. We have changed the term "quantiles" to "quantile intervals" throughout the manuscript.

2. Figure 3 Legend: "Histograms show the frequency of data." Please provide a more detailed description for the reported frequencies.

We have provided more details regarding the histograms in the legend.

3. Supplementary Figure S6: The association between Delta Meth and relative frequency of alteration is not present in all tumors, in particular for the native levels 0.75-1.0 there is one tumor for which the association has the opposite direction. Is there any particular reason for this?

We agree that the associations that occur in the opposite direction in Supplementary Figure S6 is an interesting feature. When we investigated this anomaly, we found that this cancer type (CHOL; cholangiocarcinoma) had relatively few alterations at the upper quantile interval in comparison to the other cancer types and other quantile intervals. We have concluded that this is a unique feature of CHOL and not an error in the analysis.

4. Lines 335-336: "We considered the cancer tissue DNA methylation level to be aberrant if it fell outside of the native DNA methylation level.". Please provide a more specific definition.

We would like to apologize for the lack of clarity regarding this point, and we have provided a more specific definition in the Methods section. (Lines 402-403).

5. Lines 336-337: "We used the equation (aberrant cancer DNA methylation level – native DNA unmethylated level) and (aberrant cancer DNA methylation level – native DNA methylation level) to calculate the Δ Meth in

the hypomethylation and hypermethylation direction respectively”. Shouldn’t it be the opposite for the hypomethylation direction (i.e. native DNA unmethylated level - aberrant cancer DNA methylation level) ?

Correct, this is a typo. We have corrected this error in the manuscript.

6. Lines 346-347: Please provide more details on the aggregation of sites based on the relative frequency of aberrant events and, in particular, on how the grouping variable is defined.

We apologize for the lack of clarity and have included a more detailed description describing the aggregation calculation in the Methods section. (Lines 413-417).

7. Supplementary figures S7-S8 legend: “in the hypomethylation direction a measured”. Typo here.

We would like to thank the reviewer for pointing out this typo and we have corrected this error in the manuscript.

Reviewer #2 (Remarks to the Author):

Downs et al. propose a new metric for quantifying changes in DNA methylation. Instead of the standard ΔMeth (tumour – normal, or treatment – control), they introduce relative ΔMeth , which divides ΔMeth by the baseline methylated level when the change is a loss (hypomethylation) or by the baseline unmethylated level when the change is a gain (hypermethylation).

Using relative ΔMeth , the authors show that methylation changes appear more consistent across CpGs with different starting values, whereas absolute ΔMeth is strongly influenced by the native level. They argue that relative ΔMeth is a less biased way to detect biologically meaningful alterations. Gene-set enrichment analysis (GSEA) of the CpGs with the highest relative ΔMeth in TCGA tumours highlights cancer-specific pathways, whereas the same analysis based on absolute ΔMeth points to broader, less disease-specific pathways.

The core idea of treating the mean relative ΔMeth as background and identifying loci whose relative ΔMeth exceeds that background is potentially important.

However, we believe several key issues need to be addressed (details below).

We would like to thank Reviewer 2 for their insightful suggestions to better understand the relative ΔMeth approach. To address these comments and suggestions, we have added 3 new paragraphs in the Results section, 2 new paragraphs in the Discussion section, new Figure 6 and 7, new Supplemental Figure S9 and S15, and new Supplemental Tables S13-21. Overall, we believe that these new results have dramatically improved the scientific significance of our manuscript.

1 Global versus regional change

A central assertion—“DNA-methylation alterations occur as a global rate of change relative to the native level”—is problematic. Although the net gain or loss is, on average, proportional to the starting level, the apparent consistency arises from averaging highly heterogeneous effects with distinct biological implications. Numerous studies have shown that DNA-methylation changes cluster in specific compartments: e.g. in cancer, focal island hypermethylation and large-scale hypomethylation tend to occur in late-replicating, lamina-associated regions (also known as PMDs; see doi: 10.1038/nsmb.2518, 10.1038/nrm.2015.31, 10.1038/ng.969). Hypomethylation of PMDs, often age-associated, also contributes to tumorigenesis (doi: 10.1038/s41588-018-0073-4). Averaging all Open-Sea probes therefore yields a figure that does not represent either highly methylated domains (HMDs) or PMDs.

Indeed, the first experiment presented, 5-AZA treatment, may be the only one that produces a true genome-wide loss: under 5-AZA, virtually every Open-Sea CpG loses methylation by a similar relative ΔMeth , whereas in cancer only the Open-Sea CpGs within PMDs lose methylation.

Specific recommendations

- a) Acknowledge prior evidence that methylation loss is compartment-specific, not uniform.
- b) Re-evaluate ΔMeth and relative ΔMeth separately within late-replicating / lamina-associated domains (or PMDs / hypomethylated blocks) to test whether the new metric captures biologically meaningful changes there.

We would like to thank the reviewer for raising this insightful point, and we have acknowledged information regarding the loss of DNA methylation in PMDs and HMDs in our manuscript.

In our revised manuscript, we further investigated large ΔMeth and relative ΔMeth hypomethylation alterations that occur in common PMDs, common HMDs and in neither PMDs/HMDs. We found that both the CpG sites selected using large ΔMeth and relative ΔMeth were mostly in PMDs (Figure 7A). Interestingly, we found that relative ΔMeth selected for more CpG sites in common HMDs and in neither PMDs/HMDs in comparison to ΔMeth (Figure 7A).

Similar to our previous findings, we found that the large ΔMeth in common PMDs, common HMDs, and neither PMDs/HMDs enriched for genes in brain pathways (Figure 7B and Supplemental Table S16-18). Conversely, we found that CpG sites with large relative ΔMeth in common HMDs and neither PMDs/HMDs domains were enriched for genes in pathways with known importance for tumor development in comparison to those in common PMDs (Figure 7C and Supplemental Table S16-18).

From these results, we confirmed the previous findings that alterations at different genomic compartments have different biological functions for tumor development. Moreover, we found that these biological differences are more obvious when using the relative Δ Meth approach in comparison to the Δ Meth approach.

Details describing these results can be found in the Results section lines 294-314 and Discussion section line 355-367).

2 Biological relevance of the selected CpGs

The authors claim that relative Δ Meth identifies more relevant CpGs, based on GSEA of the top 10 % most variable sites. This apparent improvement likely reflects the inclusion of low-baseline hypomethylated CpGs and high-baseline hypermethylated CpGs. Demonstrating functional importance requires further evidence, e.g. by correlating methylation and expression specifically for this subset.

We have provided new correlation plots showing DNA methylation levels and expression levels for the top two CpG sites that had the strongest correlation at different native DNA methylation quantile intervals (Supplementary Figure S12 and S13).

We found that in the hypomethylation direction, CpG sites with large relative Δ Meth in the lowest quantile interval show a more even spread of DNA methylation levels across the beta scale in comparison to the CpG sites with large Δ Meth, which show large clusters of DNA methylation levels at large beta values.

When we investigated the CpG sites with large relative Δ Meth in the highest quantile interval in the hypermethylation direction, we found very little expression change between the DNA methylation levels 0.0-0.6. In our experience, this feature is uncommon in Δ Meth-selected hypermethylation alterations and suggests that relative Δ Meth may be revealing new and interesting DNA methylation biology.

3 Comparison with existing alternatives

An established alternative to Δ Meth is Δ M, where $M = \text{logit}(\beta)$. Is relative Δ Meth superior to Δ M for highlighting biologically meaningful sites? A direct comparison would strengthen the paper.

We would like to thank the reviewer for the suggestion to investigate whether the $M = \text{logit}(\beta)$ approach would also select for biologically meaningful DNA methylation alterations. The Δ M-value is a good approach to compare since it increases the resolution of DNA methylation loss when the levels are near 0.0, which can lead to the selection of CpG sites with low native levels.

As expected, we found that large Δ M-value hypomethylation alterations occurred at CpG sites in all native level quantile intervals (Figure 6A and Supplementary Figure S15). However, when we performed GSEA on the 10% largest Δ M-values that occurred in the hypomethylation direction, we found that the selected genes were enriched in pathways for the midbrain (Figure 6C and Supplementary Table S13 and S15). Again, we found that the unique relative Δ Meth selected genes were enriched in pathways known to be important for tumor development (Figure 6C and Supplementary Table S14). Thus, we conclude that the ability of relative Δ Meth to enrich for biologically significant alterations in the hypomethylation direction is not simply due to the selection of CpG sites with low native levels.

Details describing these results can be found in the Results section lines 274-293 and Discussion section line 343-354).

4 Minor points

a) Baseline definition. The “native” baselines appear to pool all normal tissues. Because DNA methylation is strongly tissue-specific, this will preferentially select the small subset of CpGs that are constitutively high or low in every tissue. Using matched normal tissue would be more appropriate.

We agree with the reviewer that using matched tissues would provide more biologically important information regarding DNA methylation alterations in comparison to using native levels derived from a diverse dataset of normal tissues. Although it is uncommon to have matched tissues, there are a number of matched normal-tumor tissues in the TCGA database. To investigate whether using matched tissues

would cause a large difference in the relative Δ Meth distribution profiles, we calculated the native levels and relative Δ Meth using only matched breast tissues (n=97). We found that matched breast tissues had similar distribution profiles in comparison to the breast cancer relative Δ Meth distribution profiles made using the native levels calculated from the pooled normal tissues, see plots below.

While the overall conclusions regarding relative Δ Meth may hold true, we agree with the reviewer that more biological information can be determined by the investigation of match tumor-normal tissues. In our next study, we plan to investigate relative Δ Meth alterations in matched tissues.

b) CpG-to-gene mapping. For DNAm–expression correlations, how is per-gene methylation derived? Promoter methylation has very different effects from gene-body methylation; the analysis should stratify by genomic context.

We derived the per-gene methylation information from the array annotation file from the Illumina website. This information is now included in the Methods section. (Lines 455-456).

We agree that promoter methylation has different effects from gene-body methylation. To this point, we have included the genomic context in relation to the CpG island of the top 10% most frequently altered CpG sites in Supplementary Tables S3 and S4 and included a new Supplemental Figure S9. Although the biological functions of different genomic regions are important, we feel it is slightly outside the focus of this manuscript and deserves its own investigation.

In our next study, we plan to investigate whether the relative Δ Meth approach can help illuminate new DNA methylation biological features in tumors. To accomplish this, we will investigate the different tumor types independently. We will also stratify the CpG sites based on their genomic regions within the genes and will continue to investigate PMD and HMD genomic compartments.

RESPONSE TO REVIEWERS' COMMENTS

Reviewer #2 (Remarks to the Author):

The authors have partially addressed my concerns from the original review. In particular, I appreciate the work done in response to the issue of methylation patterns (both normal/baseline and 'aberrant') being very different between different genomic regions. My remaining concern is that the new approach of studying changes in genomic regions defined as HMDs vs PMDs for all cancers grouped together is difficult to interpret since genomic methylation patterns are so tissue specific. (In fact a list of HMDs and PMDs for a given sample is a great predictor of which tissue type the sample comes from). In my opinion, this analysis would have to be done in a tissue-specific manner (or at least a subset of individual tissues) to be meaningful. (I.e. define HMDs, PMDs for that tissue, and then relate it to methylation alterations in the same tissue).

We would like to thank the reviewer for their thoughtful suggestion to reanalyze the genomic compartment analysis in a tissue-specific manner. In our revised manuscript, we investigated large Δ Meth and relative Δ Meth hypomethylation alterations that occur in common PMDs, common HMDs and in neither PMDs/HMDs in breast and colorectal cancers.

To accomplish this, we first identified the CpG sites of the array that were concordant with the previously identified common genomic compartments. We defined the CpG sites to be concordant if the median DNA methylation level of the normal-adjacent breast and colorectal tissues were $> 60\%$ in common HMDs, between 30% and 60% in common PMDs, and $< 30\%$ in neither PMDs/HMDs.

Using the concordant CpG sites, we then identified the top 10% largest Δ Meth and relative Δ Meth alterations in the cancers. We found that in both tissue types, the CpG sites selected using large Δ Meth were mostly in HMDs while the large relative Δ Meth were mostly in PMDs (Figure 7A and B). Similar to our previous findings, we found that the large Δ Meth in common HMDs enriched for genes in pathways confounding for tumor development (Figure 7C and E and Supplemental Tables S16 and S19). Conversely, we found that CpG sites with large relative Δ Meth in all genomic compartments were enriched for genes in pathways with known importance for tumor development (Figure 7D and F and Supplemental Tables S17, S18, and S20-22). Furthermore, we also found that in both breast and colorectal cancers, alterations in HMDs were enriched in more immune pathways than alterations in PMDs and neither PMDs/HMDs.

From these results, we confirmed the previous findings that alterations at different genomic compartments have different biological functions for tumor development. Moreover, we found that these biological differences are more obvious when using the relative Δ Meth approach in comparison to the Δ Meth approach.